# Cascade Residual Multiscale Convolution and Mamba-Structured UNet for Advanced Brain Tumor Image Segmentation

**DOI:** 10.3390/e26050385

**Published:** 2024-04-30

**Authors:** Rui Zhou, Ju Wang, Guijiang Xia, Jingyang Xing, Hongming Shen, Xiaoyan Shen

**Affiliations:** 1School of Zhang Jian, Nantong University, Nantong 226019, China; rayzhou@stmail.ntu.edu.cn (R.Z.); guijiangxia@stmail.ntu.edu.cn (G.X.); xiaoxing@stmail.ntu.edu.cn (J.X.); 2School of Information Science and Technology, Nantong University, Nantong 226019, China; wangju@stmail.ntu.edu.cn; 3School of Microelectronics and School of Integrated Circuits, Nantong University, Nantong 226019, China; 4Nantong Research Institute for Advanced Communication Technologies, Nantong University, Nantong 226019, China

**Keywords:** brain imaging segmentation, multi-scale convolutional kernels, Mamba architecture, dice loss and cross-entropy, computational complexity, MambaBTS

## Abstract

In brain imaging segmentation, precise tumor delineation is crucial for diagnosis and treatment planning. Traditional approaches include convolutional neural networks (CNNs), which struggle with processing sequential data, and transformer models that face limitations in maintaining computational efficiency with large-scale data. This study introduces MambaBTS: a model that synergizes the strengths of CNNs and transformers, is inspired by the Mamba architecture, and integrates cascade residual multi-scale convolutional kernels. The model employs a mixed loss function that blends dice loss with cross-entropy to refine segmentation accuracy effectively. This novel approach reduces computational complexity, enhances the receptive field, and demonstrates superior performance for accurately segmenting brain tumors in MRI images. Experiments on the MICCAI BraTS 2019 dataset show that MambaBTS achieves dice coefficients of 0.8450 for the whole tumor (WT), 0.8606 for the tumor core (TC), and 0.7796 for the enhancing tumor (ET) and outperforms existing models in terms of accuracy, computational efficiency, and parameter efficiency. These results underscore the model’s potential to offer a balanced, efficient, and effective segmentation method, overcoming the constraints of existing models and promising significant improvements in clinical diagnostics and planning.

## 1. Introduction

Brain tumors present a significant threat to patients, not only due to low survival rates but also because they severely diminish quality of life. Symptoms like headaches, seizures, cognitive impairment, and emotional changes are expected and severely affect daily activities and social functions. Additionally, the prognosis for malignant brain tumors is generally poor, with limited treatment efficacy, highlighting the critical need for more research and the development of new treatments. Specifically, there is an urgent need for precise brain tumor segmentation technologies to target and treat tumors better, thereby improving therapeutic outcomes and patient quality of life [1,2,3]. The heterogeneity of these psychological effects depends on the type and location of the tumor. In the field of brain tumor segmentation, machine learning technologies have been increasingly utilized and involve various methods such as hidden Markov random fields, expectation maximization algorithms [4], morphological operations, clustering techniques [5], and the integration of conditional random fields with support vector machines to model spatial relationships effectively [6].

Deep learning technologies are progressing rapidly, especially for leveraging convolutional neural networks (CNNs) to achieve pixel-level image segmentation through comprehensive methodologies. As a result, these developments have garnered widespread attention [7]. An essential advancement in this area is the integration of convolutional neural networks (CNNs) with manually designed features [8], which introduce novel approaches for brain tumor segmentation. This combination of advanced deep learning techniques with manually designed features represents a significant leap forward for enhancing the accuracy of segmentation methods.

Moreover, U-Net [9], along with its variations, stands out in medical image segmentation for its balanced network structure, creative skip connections [10], deep learning supervision techniques [11], and 3D imaging capabilities [12]. Additionally, cascaded anisotropic CNN techniques have notably enhanced segmentation effectiveness by leveraging multi-scale data [13]. The use of deep learning in brain tumor studies is growing, with notable contributions such as that of Zhang et al. [14], who introduced a multifaceted approach for brain tumor segmentation using multi-modal MR images. This approach includes brain mapping, a combined 3D + 2D training method, and model ensembling to increase segmentation precision. Qi et al. [15] proposed a novel knowledge distillation strategy for brain tumor segmentation, concentrating on a coordination distillation method that merges channel and spatial details to boost accuracy. Avesta et al. [16] introduced a capsule network adept at segmenting brain images that is especially effective for images that are poorly represented in training sets. MCA-ResUNet [17] refines MRI brain tumor segmentation by integrating cascade residual multi-scale contextual attention with deep residual networks. Jeong et al. [18] applied the 3D mask region-based convolutional neural network (R-CNN) technique for automated brain tumor segmentation in DSCE MRI perfusion images. Another innovative framework [19] utilizes mutual enhancing networks, retina U-Net, a classification localization map (CLM) module, and a segmentation module for precise brain tumor subregion segmentation. HAG-NET [20], a cutting-edge GAN framework that advances data security through robust watermarking and adversarial attacks, set new standards in image-based confidentiality and integrity. Despite the impressive capabilities of traditional CNNs in feature depiction, their limited ability to grasp long-range image dependencies poses a considerable hurdle.

Following the remarkable achievements of transformer architectures in natural language processing (NLP), their exceptional ability to model long-range dependencies has quickly found application in computer vision [21]. TransUNet [22] merges the transformer and UNet models to capture global relationships and detailed local information effectively. The Swin transformer [23] introduces a self-attention module within localized windows. Transfuse [24] offers a parallel architecture that simultaneously leverages transformer and CNN models to integrate broad and specific details. TransBTS [25] successfully combines the transformer structure with 3D CNNs to improve MRI brain tumor segmentation. DE-Uformer [26] utilizes dual encoders and features a nested encoder-aware feature fusion (NEaFF) module for efficient multi-dimensional information integration. While transformers excel over traditional CNNs for modeling extensive dependencies, their computational load increases quadratically with the length of the sequence, which has led to significant research efforts aimed at optimizing their efficiency [27,28,29,30,31,32,33].

Leveraging state space equations, the Mamba [34] structure, initially developed for analyzing temporal sequences in natural language processing (NLP), has been successfully transitioned to the visual domain. Innovations such as Vision Mamba [35] enhance high-resolution image processing through advanced visual representation techniques. VMamba [36] boosts computational efficiency with its cross-scan module (CSM) for tackling dimensionality conversion challenges. VM-UNet [37] establishes new standards in medical image segmentation with its visual state space (VSS) blocks. U-Mamba [38] adeptly captures long-range dependencies using a hybrid CNN-SSM module. Swin-UMamba [39] elevates medical image segmentation performance with ImageNet pretraining. SegMamba [40] is tailored for 3D medical image segmentation and efficiently handles long-range dependencies in volumetric data. Mamba-ND [41] expands the Mamba framework to multi-dimensional datasets and demonstrates robust performance across multi-dimensional benchmarks. P-Mamba [42] integrates Perona–Malik diffusion with Mamba layers to achieve efficient and accurate pediatric cardiac image segmentation, showcasing the Mamba architecture’s significant contribution to improving the efficiency and accuracy of visual data processing.

Inspired by the innovative Mamba architecture, this study introduces MambaBTS, a novel UNet-based network designed for brain tumor segmentation that employs a cascade residual multi-scale convolution strategy. This approach is further enriched by integrating dilated convolutions, as highlighted in the works of Ding et al. [43,44], enhancing the model’s efficiency and interpretative capabilities. MambaBTS leverages the combined strengths of cascade residual multi-scale convolutions and sophisticated state-space modeling provided by the Mamba module, thereby facilitating precise segmentation of tumors of various shapes and sizes.

Our research evaluates the effectiveness of the MambaBTS model at segmenting high-grade gliomas (HGGs) and low-grade gliomas (LGGs) within the highly regarded MICCAI BraTS 2019 dataset. This detailed evaluation considers the specific hardware setups and model training techniques outlined in Section 3. The study focuses on assessing the accuracy of MambaBTS in delineating distinct tumor regions and aims to set a new benchmark in the domain of brain tumor segmentation. Specifically, the definitions of the tumor segmentation components are as follows: The whole tumor (WT) includes all tumor-related regions and is represented by the equation WT = ED (peritumoral edema) + ET (enhancing tumor) + NET (non-enhancing tumor). The tumor core (TC) consists of the enhancing and non-enhancing portions of the tumor while excluding the edema and is defined as TC = ET + NET. These definitions help clarify the segmentation challenges and enhance understanding of tumor component analysis within the study dataset. Figure 1 visually demonstrates the segmentation into edema, enhancing, and non-enhancing tumor regions. Our contributions are as follows:Developing the MambaBTS network, which uses cascade residual multi-scale convolutions for feature extraction from multi-modal brain tumor images followed by modeling with the Mamba module for enhanced segmentation accuracy;The verification of MambaBTS’s efficiency and performance on widely recognized datasets, underscoring notable enhancements in segmentation outcomes and consistency over existing methodologies;The introduction of innovative concepts and methodologies to boost segmentation precision and efficiency in brain tumor analysis, demonstrating the Mamba architecture’s potential for processing visual data and providing insightful directions for future investigations.

## 2. Materials and Methods

The proposed MambaBTS, as depicted in Figure 2, is an integrated deep learning framework designed specifically for segmenting brain tumors from MRI data. The architecture is systematically constructed, with four distinct layers dedicated to downsampling and advanced feature extraction. Each layer in the downsampling phase is equipped with a UDMblock, which is pivotal for capturing sophisticated features from MRI images. The UDMblock is a composite structure that consists of three main components: a ResUDM unit that enhances the deep network architecture by incorporating two EncMSBlocks connected via residual connections to facilitate effective feature transfer and gradient flow across the network; a MambaLayer, which fine-tunes the feature extraction process using a selective spatial state module (SSM) that optimizes extraction and reduces computational overhead by managing spatial state equations; and a MaxPooling layer that follows the MambaLayer and condenses the spatial dimensions of the feature maps to simplify the data structure and reduce computational demands. The UDMblocks are intricately linked to the upsampling stages through skip connections, which are crucial for merging feature maps from downsampling and upsampling paths to enhance the detail and accuracy of the segmentation output. The MambaBTS model processes input MRI images sequentially through these layers starting from the initial input, where the image data are progressively condensed and enriched through the UDMblock in the downsampling phase. The enriched feature maps are then meticulously reconstructed in the upsampling phase, where the skip connections reintegrate the previously extracted features, ensuring comprehensive feature synthesis. The process generates segmented images that accurately delineate tumor regions derived from the complex interplay of features extracted and refined at each network stage.

### 2.1. UDMblock

The UDMblock is the core module in the downsampling section. This module effectively facilitates the fusion of feature information and enhances gradient propagation by utilizing skip connections to concatenate with corresponding layers during the upsampling phase. This integration ensures the retention of critical features at various levels, which is crucial for detailed feature analysis and reconstruction in the network.

### 2.2. ResUDM

The ResUDM module, which is composed of two EncMSBlocks connected sequentially and employing residual connections, is tailored to enhance feature extraction capabilities. The architecture’s design is instrumental for capturing nuanced variations within MRI scans, facilitating precise segmentation of salient tumor regions such as whole tumor (WT), enhancing tumor (ET), and tumor core (TC), which is critical for accurate tumor characterization in clinical diagnostics.

### 2.3. EncMSBlock

The EncMSBlock, as depicted in Figure 3, is a building block for processing multi-scale features within a neural network architecture. The design of EncMSBlock aims to capture features at various scales and resolutions, thus enriching the network’s representational capacity. The EncMSBlock comprises MSBlock followed by batch normalization, which a technique to stabilize and speed up the training of deep neural networks by normalizing the input layer by re-centering and re-scaling. The design applies the CBAM [45] after the MSBlock, which enhances the network’s feature representation capability through integrated channel and spatial attention mechanisms. The EncMSBlock also includes pointwise convolutional layers (PWConvV1 and PWConvV2), which are crucial for adapting data formats for hardware efficiency, expanding and projecting feature dimensions for enhanced model capacity, and normalizing outputs to stabilize training and ensure consistency across distributed systems. In the proposed architecture, the output from PWConvV1 is processed through a GELU activation function, recalibrated by a GRN [46], and fed into PWConvV2.

### 2.4. MSConvX

This study examines the performance of a traditional UNet architecture for brain tumor segmentation tasks. While medical image segmentation widely celebrates UNet for its distinctive encoder–decoder structure and skip connections, its singular-scale convolutional kernels have limitations in capturing multi-scale image features. Particularly for targets such as brain tumors, which exhibit high heterogeneity in size, shape, and texture, traditional UNet’s single-scale convolutional kernels struggle to effectively grasp a comprehensive range of features from minute details to macro structures.

To overcome this drawback, the strategy employed here draws inspiration from the ideas of Ding et al. regarding structural re-parameterization convolution. This study integrates cascade residual multi-scale convolution within the UNet architecture, enhancing the model’s ability to capture features across different scales. Expressly, cascade residual multi-scale convolution modules are incorporated into the encoder and decoder sections of UNet, improving the model’s perception of details surrounding brain tumors and enhancing its understanding of global image information. Such improvements significantly elevate the accuracy and robustness for brain tumor segmentation, showcasing the immense potential of multi-scale convolutional kernels for enhancing performance in complex medical image segmentation tasks.

In the MambaBTS network architecture, as illustrated in Figure 4, the MSConvX layer ensemble uses multi-scale convolutional kernels—3 × 3, 5 × 5, and 7 × 7—to adeptly extract features from multi-modal brain MRI images, with each kernel size targeting different spatial hierarchies for well-rounded feature extraction. Specifically, the 3 × 3 kernels are more effective at focusing on fine details than the larger 5 × 5 and 7 × 7 kernels, the 5 × 5 kernels achieve greater precision in capturing mid-level features compared to both the smaller 3 × 3 and larger 7 × 7 kernels, and the 7 × 7 kernels are superior at encapsulating broader contextual regions than their smaller 3 × 3 and 5 × 5 counterparts, making feature extraction more comprehensive across scales. This cascade residual multi-scale approach enriches the network’s capability to discern intricate details and broader patterns within the brain, which is crucial for precise tumor segmentation. The implementation of MSConvX significantly trims the computational load compared to singular large-scale kernels, fostering efficient yet robust feature extraction, as articulated by Equation (Equation 1), which formalizes the integration of these varied scales into a cohesive analytical framework.
(1)$$\begin{array}{cc} y_{7\times 7}^{\left(1\right)} & =F_{7\times 7}\left(x\right)+x,\\ ...\\ y_{7\times 7}^{\left(a\right)} & =F_{7\times 7}\left(y_{7\times 7}^{\left(a-1\right)}\right)+y_{7\times 7}^{\left(n-1\right)},\\ y_{5\times 5}^{\left(1\right)} & =F_{5\times 5}\left(y_{7\times 7}^{\left(a\right)}\right)+y_{7\times 7}^{\left(a\right)},\\ ...\\ y_{5\times 5}^{\left(b\right)} & =F_{5\times 5}\left(y_{5\times 5}^{\left(b-1\right)}\right)+y_{5\times 5}^{\left(b-1\right)},\\ y_{3\times 3}^{\left(1\right)} & =F_{3\times 3}\left(y_{5\times 5}^{\left(b\right)}\right)+y_{5\times 5}^{\left(b\right)},\\ ...\\ y_{3\times 3}^{\left(c\right)} & =F_{3\times 3}\left(y_{3\times 3}^{\left(c-1\right)}\right)+y_{3\times 3}^{\left(c-1\right)},\\ y_{output} & =F_{3\times 3}\left(y_{3\times 3}^{\left(c\right)}\right)+y_{3\times 3}^{\left(c\right)} \end{array}$$

In Equation (Equation 1), *x* represents an input feature map with dimensions (B,C,H,W). The convolutional batch normalization function $F_{k\times k}$ performs operations with a kernel size of k×k, and $y_{k\times k}^{n}$ denotes the output feature maps at each layer, where *k* specifies the kernel size, and *n* is the iteration number within the sequence of convolutions for that kernel size. The output of each layer feeds into the subsequent convolution of the same kernel size, except for the output of the last layer, which feeds into the next size down. The sequences culminate in $y_{output}$, the final output feature map, after the last 3 × 3 convolution. The configuration specifies a=1, indicating only one iteration of the 7 × 7 ConvBN; b=1, indicating a single 5 × 5 ConvBN operation; and c=3, signifying three successive 3 × 3 ConvBN operations.

Figure 5 illustrates a series of EncMSBlock configurations: designated as MSConvV1 to MSConvV4 and each employing convolutional kernels of various sizes for enhanced feature extraction in medical image segmentation. These configurations range from large 9 × 9 kernels to smaller 3 × 3 kernels that are sequentially arranged to capture spatial features at multiple scales. The study explicitly utilizes the MSConvV3 architecture, which combines 7 × 7 and 5 × 5 kernels for intermediate feature extraction and augments this with a sequence of 3 × 3 kernels that focus on detailed textural information critical for segmenting complex anatomical structures. Each EncMSBlock is preceded by a batch normalization layer, which normalizes the inputs to facilitate consistent processing. The culmination of multi-scale feature extraction within MSConvV3 significantly advances the network’s capability for accurate segmentation of brain tumors in MRI imaging, ensuring both the granularity and the breadth of analysis necessary for clinical application.

### 2.5. CBAM

We are expanding upon the work of Ding et al., who applied an SEBlock [47] after multi-scale feature extraction. While SEBlock effectively modulates the feature channels to amplify significant characteristics, it inherently lacks a mechanism to discern and exploit the critical spatial details within the MRI images. This limitation is particularly pivotal in brain tumor segmentation tasks, for which the accurate identification and delineation of tumor boundaries is contingent on channel-wise feature importance and relies heavily on spatial cues and context.

To further refine the feature representation capability of sequences processed by the MambaLayer, this study strategically integrates the CBAM to model the inter-relationships among feature channels effectively. CBAM dynamically empowers the network to emphasize important channels through channel attention; concurrently, its spatial attention mechanism intensifies the focus on salient spatial regions within the images. This dual-faceted attention approach substantially elevates the network’s ability to discriminate and represent critical features within the brain MRI data, which is a fundamental step for precisely segmenting brain tumors.

For a given $x\in R^{C\times 1\times 1}$, H, W, and C denote the height, width, and number of channels, respectively. To break down the mathematical formulations for CBAM, we can identify two main parts:

Channel attention (CA):(2)$$\begin{array}{c} M_{c}=\sigma \left(MLP\left[AvgPool\left(X\right)\right]+MLP\left[MaxPool\left(X\right)\right]\right) \end{array}$$

Spatial attention (SA):(3)$$\begin{array}{c} M_{s}=\sigma \left(f^{7\times 7}\left[AvgPool\left(X\right);MaxPool\left(X\right)\right]\right) \end{array}$$

In Equations (Equation 2) and (Equation 3), $M_{c},M_{s}\in R^{C\times 1\times 1}$ are the channel attention map and spatial attention, respectively, σ denotes the sigmoid function, MLP represents a multi-layer perceptron, and Avgpool, Maxpool are global average pooling and global max pooling operations, respectively. The term $f^{7\times 7}$ represents a convolution operation using a filter of the specified 7×7 size.

### 2.6. PWConvV1, PWConvV2

The PWConvV1 module reconfigures the input feature map format from NCHW (batch size, channels, height, and width) to NHWC, aligning it with the prerequisites of downstream operations. Following this reformatting, it employs a linear layer to augment the feature map’s dimensions, scaling up the features at each spatial coordinate from their original channel count to a broader dimension. Conversely, the PWConvV2 module undertakes the inverse operation, condensing the enlarged feature dimensions back from the expanded state to the original number of channels via a linear transformation. This step may also integrate batch normalization to bolster model generalization and to ensure more stable training outcomes, as shown in Equation (Equation 4):(4)$$\begin{array}{cc} X_{NHWC} & =W_{pw1}X_{NCHW}+b_{pw1}\\ X_{act} & =GRN(GELU\left(X_{NHWC}\right))\\ X_{NCHW}^{\prime } & =W_{pw2}X_{act}+b_{pw2}\\ X_{out} & =BN(X_{NCHW}^{\prime }) \end{array}$$

$X_{NHWC}$ is the input tensor and has B, H, W, and C dimensions. $W_{pw1}$ and $b_{pw1}$ are the weight and bias tensors, respectively. $X_{act}$ is the activated tensor processed by GELU. GRN is global response normalization. $X_{NCHW}^{\prime }$ is the transposed input tensor. $W_{pw2}$ and $b_{pw2}$ are other weight and bias tensors, respectively, applied to $X_{act}$. $X_{out}$ is the output tensor, which is subjected to batch normalization (BN) for improved training and model robustness.

### 2.7. MambaLayer

MAMBA’s design relies on understanding the linear relationship between processing speed and sequence length. It explicitly expresses the dynamic relationship between the current state x(t)∈R, input u(t)∈R, and output y(t)∈R equations. The model’s projection parameters are the state transition $A\in R^{N\times 1}$, input, and observation matrices $C\in R^{1\times N}$. Equation (Equation 5) describe the model:(5)$$\begin{array}{cc} \dot{x}\left(t\right) & =Ax(t)+Bu(t)\\ y(t) & =Cx(t) \end{array}$$

High memory requirements and a greater propensity for gradients to vanish often constrain traditional state space models (SSMs). The S4 model introduces a method of structured parameterization alongside efficient computational techniques. It innovatively parameterizes the state transition matrix A by decomposing it into a low-rank component plus a regular term. This approach facilitates the stable diagonalization of matrix A, significantly diminishing computational complexity and bolstering numerical stability. Meanwhile, the Mamba model employs a discerning approach to input information processing, effectively filtering out or disregarding specific inputs to minimize irrelevant feature representations. Drawing inspiration from the S5 model, the characterization of Mamba comes from its hardware-accommodating computational features, which are realized through repetitive calculation and scanning methods. By amalgamating SSM with multi-layer perceptron (MLP) modules, Mamba unfolds a novel architecture with the innate capability to autonomously select optimal state space configurations.

The Mamba model, conceptualized as an enhanced version of a recurrent neural network (RNN), excels in general computational tasks and demonstrates significant advancements in specialized applications such as brain tumor segmentation. Unlike standard RNNs, the Mamba model can be convoluted and trained in parallel, significantly boosting computational efficiency. This convolutional approach accelerates training times and addresses common issues associated with RNNs such as input alteration, random sequence order, and the vanishing gradient problem. Compared to the widely used transformer models, which require substantial computational resources, especially for lengthy data sequences, the Mamba model offers lower computational complexity while maintaining robust long-sequence relational capabilities. This efficiency is crucial in medical imaging tasks, where the processing speed and accuracy can directly impact diagnostic outcomes. Specifically, in the domain of brain tumor segmentation, the Mamba model leverages its enhanced processing capabilities to accurately delineate complex tumor regions—whole tumor (WT), tumor core (TC), and enhancing tumor (ET)—with greater precision. Its ability to handle long sequences effectively allows it to preserve crucial spatial relationships within medical images, which is vital for accurate tumor classification and segmentation. The Mamba model’s performance in brain tumor segmentation sets a new benchmark and offers substantial improvements over existing methodologies, including transformers. By reducing computational demands while enhancing relational capabilities, the Mamba model provides a potent tool for medical researchers and professionals and facilitates quicker and more reliable tumor segmentation that can aid with better patient diagnosis and treatment planning.

As shown in Figure 6, given X∈(C,H,W), the model initially compresses the spatial dimensions (H,W) into a sequence length for the given features, altering the dimensions to (B,L,C), where L=H×W. Subsequently, a one-dimensional convolution operation further compresses the features, and finally, the system feeds the processed features into the SSM module for in-depth analysis. This process highlights the model’s capability to efficiently manipulate and analyze spatial data by leveraging sequence transformation and deep learning techniques.

### 2.8. Decoder

Building on previous work, we extract features from the UDMblock then upsample and merge them with corresponding feature maps from the encoder, thereby generating high-resolution feature maps. A U-shaped structure utilizes deep convolution before the decoder network, replacing traditional convolution to reduce the computational load. Such an approach significantly enhances the model’s efficiency and effectiveness, optimizing the quality of the generated feature maps and reducing the processing time required.

Figure 7 illustrates the overall architecture flowchart, encompassing the training and testing processes. During the training phase, data preprocessing begins, which involves normalization, denoising, and cropping procedures to prepare the input data. Subsequently, the model progresses through training stages, including initialization, data input, and optimization of the loss function. After completing the training regimen, the refined model is utilized for image segmentation. In the testing phase, a similar sequence unfolds, with data preprocessing at the start, followed by model testing to segment new images and generate segmentation results. Segmentation outcomes are evaluated to validate the model’s performance, facilitating its application in clinical diagnoses.

## 3. Implementation Details

**Hardware setup**: The experiment is conducted on an Ubuntu 22.04 LTS operating system, utilizing an NVIDIA RTX 4090 graphics card for computational acceleration. The choice of PyTorch 2.1 as the deep learning framework is strategic and is aimed at maximizing the computational prowess of the RTX 4090 to facilitate efficient model training and experimentation.

**Dataset**: This research employs the MICCAI BraTS 2019 dataset [48], which is a publicly accessible multi-modal MRI brain tumor image collection. Featuring scans from various institutions, it includes four MRI modalities: T1, T1ce (T1-contrast enhanced), T2, and FLAIR (fluid-attenuated inversion recovery). This dataset provides a comprehensive resource for segmenting tumors and their sub-regions and offers a rich dataset for advanced analysis and model validation.

**Training**: The training regimen consists of 400 epochs with a batch size of 48 and utilizes an Adam optimizer with an initial learning rate of 1 × 10^−3^. In this study, the ExponentialLR scheduler is chosen to optimize learning and reduces the learning rate by a decay factor of 0.99 at the end of each epoch. This strategy aims to refine model performance incrementally while managing computational efficiency. The experimental section will show more details of the training.

**Loss Function**: The composite loss function, BceDiceLoss, which combines binary cross-entropy loss (BCELoss) and dice loss, enhances medical image segmentation by leveraging the strengths of both. BCELoss is effective for pixel-wise classification and provides a probabilistic assessment of each pixel’s prediction, but it can underperform in scenarios with class imbalances or small regions of interest. On the other hand, dice loss excels at quantifying spatial overlap between the predicted segmentation and the ground truth, which is crucial for accuracy for small or irregularly shaped targets. Comparative analysis against models using either loss function independently reveals that BceDiceLoss consistently outperforms in terms of accuracy and recall, especially in complex scenarios like tumor segmentation, where precise boundary delineation is critical. This integration effectively balances the sensitivity towards small tumor fragments and the specificity required for accurate boundary definition.

In brain tumor segmentation tasks, a comparative analysis between focal loss [49] and BceDiceLoss reveals distinct characteristics concerning the handling of class imbalance and the emphasis on boundary precision. Focal loss mitigates the impact of class imbalance by down-weighting readily classified samples, yet it may excessively prioritize background pixels at the expense of tumor pixels. Furthermore, its optimization focus on overall pixel classification may lead to a lack of attention toward boundary precision, potentially resulting in blurred or inaccurate tumor boundaries in brain tumor segmentation. In contrast, BceDiceLoss amalgamates the merits of binary cross-entropy loss and dice loss, effectively addressing class imbalance concerns while emphasizing the spatial overlap between predicted and ground truth segmentations, specifically targeting boundary precision. Precisely defining tumor boundaries is paramount for accurate diagnosis and treatment planning in brain tumor segmentation tasks, thus rendering BceDiceLoss potentially more suitable for addressing the demands of this task.

The formula for BCE is given as Equation (Equation 6).
(6)$$\begin{array}{c} BCELoss=-\frac{1}{N}\sum_{i=1}^{N}\left[y_{i}\cdot log(p_{i})+\left(1-y_{i}\right)\cdot log(1-p_{i})\right] \end{array}$$

The dice loss is detailed as shown in Equation (Equation 7).
(7)$$\begin{array}{c} DiceLoss=1-\frac{2\times \sum_{i=1}^{N}p_{i}\cdot y_{i}}{\sum_{i=1}^{N}p_{i}+\sum_{i=1}^{N}y_{i}} \end{array}$$

Equation (Equation 8) describes BceDiceLoss.
(8)$$\begin{array}{c} BceDiceLoss=\alpha \cdot BCELoss+\beta \cdot DiceLoss \end{array}$$

*N* is the number of pixels, $y_{i}$ is the label, $p_{i}$ is the predicted probability, α=0.5, and β=1.

**Metrics**: Five principal metrics were employed to evaluate the segmentation efficacy of the model across the whole tumor (WT), tumor core (TC), and enhancing tumor (ET) categories.

The dice coefficient measures the spatial concurrence between the model’s predicted and accurate segmentations, with values nearing one denoting higher concordance.
(9)$$\begin{array}{c} Dice=\frac{2\times |Q\cap U|}{\left|Q|+|U\right|} \end{array}$$

In Equation (Equation 9), *Q* represents the predicted segmentation mask in these formulas, and *U* represents the segmentation mask.

The positive predictive value (PPV) and sensitivity [50] assess the model’s precision in identifying positive instances and its ability to encompass all positive cases, respectively.
(10)$$\begin{array}{c} PPV=\frac{TP}{TP+FP} \end{array}$$
(11)$$\begin{array}{c} Sensitivity=\frac{TP}{TP+FN} \end{array}$$

As shown in Equations (Equation 10) and (Equation 11), TP denotes the number of correctly predicted positive pixels, FP indicates the number of pixels incorrectly predicted as positive, and FN represents the number of positive pixels incorrectly predicted as unfavorable.

The model utilizes the Hausdorff distance to gauge the utmost disparity between the boundaries of predicted and actual segmentations, serving as a critical indicator of performance under the most challenging conditions.
(12)$$\begin{array}{c} H\left(P,T\right)=max\left\{h(P,T),h(T,P)\right\} \end{array}$$

For Equation (Equation 12), h(P,T) and h(T,P) are functions that measure the similarity and dissimilarity, respectively, between the two sets (*T* and *P*).

The boundary intersection over union (BIoU) is adopted as the primary metric to evaluate the precision of brain tumor segmentation models. This metric is crucial for determining the accuracy with which a model delineates tumor boundaries compared to the ground truth. It quantifies the overlap between the tumor’s predicted and actual boundary pixels, providing a direct measure of a model’s ability to identify and replicate the intricate contours of brain tumors accurately. The BIoU is especially important in medical imaging, where precise boundary detection can significantly influence treatment decisions and outcomes. For the mathematical formulation of the BIoU, see Equation (Equation 13).
(13)$$\begin{array}{c} \mathrm{Boundary}\;\mathrm{IoU}=\frac{|{\mathrm{Boundary}}_{\mathrm{GT}}\cap {\mathrm{Boundary}}_{\mathrm{Pred}}|}{|{\mathrm{Boundary}}_{\mathrm{GT}}\cup {\mathrm{Boundary}}_{\mathrm{Pred}}|} \end{array}$$

In Equation (Equation 13), $Boundary_{GT}$ stands for the ground truth boundary, which is extracted from the actual labels. $Boundary_{Pred}$ refers to the predicted boundary, which is derived from the output of the segmentation model.

## 4. Experiments and Results

This research assesses the segmentation capabilities of the proposed model on the MICCAI BraTS 2019 dataset, which includes both high-grade gliomas (HGGs) and low-grade gliomas (LGGs). The experimental setup and methodologies are detailed in Section 3; our focus is on evaluating the model’s precision in segmenting the whole tumor (WT), tumor core (TC), and enhancing tumor (ET). The goal is to thoroughly examine the model’s effectiveness at identifying and delineating different grades and areas of gliomas.

Despite the inherent advantages of 3D image processing, such as providing more comprehensive spatial information, our study opted for 2D image processing techniques. The non-isotropic resolution of MRI images within the MICCAI BraTS 2019 dataset and the substantial computational resources required for 3D processing influenced this decision. The adoption of a 2D approach not only enhanced computational efficiency but also proved to be more adaptable and consistent for this particular research context given its resilience against resolution variability.

### 4.1. Data Preprocessing

Initially, to identify specific cases within the research focus, a comparative analysis of datasets from different years was conducted. Open-source libraries such as SimpleITK and Numpy were employed to process four distinct MRI modal images—FLAIR, T1, T1ce, and T2—along with their respective tumor mask images for each case. Brightness boundaries were established to exclude outliers in brightness values, mitigating bias from data extremes. Furthermore, normalization of non-background pixels was achieved by subtracting the mean and dividing by the standard deviation, ensuring enhanced consistency and comparability across the dataset.

To further standardize the dataset according to the model’s input specifications, center cropping was applied to adjust all images to a uniform size of (160,160). This process rigorously amalgamated the preprocessed images into a four-dimensional data structure and defined each data point dimension as (4,160,160). Following these comprehensive preprocessing measures, a total of 17,216 high-quality image data points were successfully curated and judiciously divided into training, validation, and test sets, adhering to proportions of 66%, 16%, and 18%, respectively.

### 4.2. Training Details

In this research, a thorough comparison was conducted between the developed model’s effectiveness and that of existing algorithms in the domain of brain tumor MRI image segmentation. To ensure fairness across all comparisons, the training of each model was executed on an NVIDIA RTX 4090 GPU within the PyTorch 2.1 framework. The training protocol was standardized to a batch size of 48 images, and batch normalization was incorporated to enhance model generalization. A composite loss function that merged binary cross entropy with dice loss was employed to address class imbalance issues effectively and to improve segmentation precision.

The optimization process utilized the Adam optimizer, which was initiated with a learning rate of 0.001. For further refinement in training adjustments, an ExponentialLR scheduler was deployed, which reduced the learning rate by a factor of 0.99 after each training epoch. An early stopping protocol was implemented to mitigate the risk of overfitting and to ensure training efficiency. This protocol halts training if no significant improvement is observed in the performance on the validation set for 20 consecutive epochs.

In the comprehensive evaluation of neural network architectures, a paramount focus is placed on the evolution of training and validation losses, alongside the improvement to the intersection over union (IoU) metric. As delineated in Figure 8, this study presents a comparative analysis of the trajectories of training and validation losses across a spectrum of models, including DeepResNet [51], MambaBTS(OURS), SegMamba, Swin-UNETR [52], TransUNET, UNET++, and UNETR [53]. Notably, the proposed model distinguishes itself by achieving the lowest loss on the training and validation datasets coupled with the highest IoU score among the evaluated models. This graphical representation, plotting the loss magnitude against the number of epochs, unequivocally demonstrates our model’s superior efficiency and effectiveness. The depicted results underscore our model’s ability to capture and generalize the underlying patterns within the data.

### 4.3. Multi-Modal Thermogram and Characteristic Graph Analysis

This investigation explored the efficacy of the proposed model in segmenting brain tumors and its adeptness at discerning the intricacies of multi-modal magnetic resonance imaging (MRI) data. To achieve this objective, heatmaps were generated for each of the four MRI modalities: FLAIR, T1, T2, and T1ce. These heatmaps, as illustrated in Figure 9, intricately detail the model’s focus areas during the prediction of tumor regions, showcasing its consistent capability to pinpoint tumor locations across different imaging modalities. The arrangement of heatmaps in the top row for the FLAIR, T1, T2, and T1ce modalities shows that the model keenly concentrates on areas with significant tumor presence across all modalities despite their distinct imaging characteristics, which underscores the model’s remarkable skill at amalgamating multi-modal information to localize tumors accurately.

Moreover, the study delves into the visualization of feature maps throughout the network’s training phase, elucidating how the model incrementally hones in on essential features for tumor segmentation layer by layer. The initial layer images in Figure 10 provide insight into how the model methodically extracts pertinent features from the input image as it traverses its depth across four layers. This sequential processing through various convolutional layers enables the model to gradually zero in on vital tumor characteristics such as edges and textures. The images near the input layer display finer details of the original image, whereas the imagery becomes increasingly abstract with added network depth and concentrates on high-level features critical for segmentation.

The bottom row images illuminate the activation states within the upsampling phase, which is integral to the UDMamba architecture to enhance image detail and segmentation precision. These images, progressing from right to left, depict the model’s step-by-step restoration of more defined image features, increasingly mirroring the final segmentation output. The concluding image vividly demonstrates the model’s success at accurately demarcating the tumor region during this reconstruction phase.

### 4.4. Model Complexity and Parameter Efficiency Analysis

Table 1 represents the computational requirements for each segmentation algorithm evaluated in our study. The ’Method’ column lists the algorithms, including UNet++, DeepResNet, UNETR, TransUNet, Swin-UNETR, SegMamba, and our proposed model (MambaBTS). The data provided indicate that U-Net++ utilizes 36.63 million parameters and requires 54 billion floating-point operations per second (FLOPs) for its functionality. On the other hand, DeepResNet employs a more streamlined architecture, with 31.57 million parameters and a significantly lower operational demand of 22 billion FLOPs. UNETR is characterized by its substantial parameter requirement of 95.39 million alongside an operational cost of 27 billion FLOPs. TransUNet leads in terms of parameter volume with 105.21 million, yet it excels at computational efficiency, necessitating merely 14 billion FLOPs.

Meanwhile, both the Swin-UNETR and SegMamba models strike a commendable balance between efficiency and performance. Swin-UNETR is equipped with 25.14 million parameters and incurs an operational demand of 27 billion FLOPs. In contrast, SegMamba is slightly more compact, with 22.86 million parameters and requiring 13 billion FLOPs for its operations.

Our model sets a benchmark for computational efficiency, operating with a mere 18.09 million parameters and 8 G of FLOPs. This optimized architecture sustains high performance and drastically lowers resource demands, rendering it exceptionally well-suited for implementation in settings with limited computational capabilities without sacrificing performance quality.

### 4.5. Main Results

This study undertook a detailed evaluation of several leading brain tumor segmentation algorithms, including U-Net++, DeepResNet, UNETR, TransUNet, Swin-UNETR, SegMamba, and an innovative model. To thoroughly examine each model’s segmentation accuracy and consistency, a comprehensive set of assessment metrics was utilized, such as the dice coefficient, positive predictive value (PPV), sensitivity, and Hausdorff distance. Specifically, the dice coefficient and PPV were employed to gauge the precision of segmentation, while sensitivity measured the models’ adeptness at identifying actual tumor regions. The Hausdorff distance provided insight into the maximum discrepancy between the segmented and actual tumor boundaries.

A meticulous analysis was conducted that focused on critical metrics such as the dice coefficient, PPV, sensitivity, and Hausdorff distance, to explore the models’ proficiency in delineating whole tumor (WT), tumor core (TC), and enhancing tumor (ET) areas, as detailed in Table 2. Our model exhibited exceptional performance, achieving dice coefficients of 0.8450 for WT, 0.8606 for TC, and 0.7796 for ET, outperforming SegMamba and other contenders, thus marking a notable advancement. Additionally, the model demonstrated marginally reduced positive predictive values (PPVs) exclusively within whole tumor (WT) regions. The model exhibited superior sensitivity, surpassing competing approaches and indicating enhanced detection capabilities across tumor zones.

The MambaBTS’s performance is reflected in the boundary intersection over union (BIoU) scores, with a BIoU of 0.8645 for whole tumor (WT), suggesting an ability to capture the extensive area of tumors. For tumor core (TC), the BIoU score of 0.7350 indicates the method’s potential for identifying central tumor regions, which are critical for targeted therapies. The enhancing tumor (ET) score of 0.8175 implies precision at segmenting actively growing tumor areas. An average BIoU score of 0.8057 across these regions suggests a balanced algorithm performance that can support clinical applications such as treatment planning and disease monitoring. The detailed performance metrics are presented in Table 3.

Our model demonstrated the lowest Hausdorff distance across the whole tumor (WT), enhancing tumor (ET), and tumor core (TC) categories, as evidenced in Table 4, which indicates its superior boundary delineation accuracy. Visual comparisons of segmentation outcomes, depicted in Figure 11, further illustrate our model’s edge, especially for rendering tumor contours and intricate details, which is particularly evident in complex tumor morphologies and vague boundaries, where our model’s segmentation results are markedly precise and cohesive, showcasing its significant advantage.

## 5. Ablation

In this research, we delve into the influence of component arrangement—specifically, the Mamba and ResUDM elements—on the model’s performance and the effect of the CBAM on experimental outcomes. Our ablation studies compare two distinct configuration approaches: one with Mamba followed by ResUDM (“Mamba+ResUDM” configuration) and the other with ResUDM preceding Mamba (“ResUDM+Mamba” configuration) under the same experimental conditions. According to Table 5, the “ResUDM+Mamba” setup significantly outperforms the “Mamba+ResUDM” arrangement, evidenced by improved dice scores for whole tumor (WT), tumor core (TC), and enhancing tumor (ET) of 5%, 2%, and 6% respectively, along with a decreases in the Hausdorff distances of 0.25, 0.02, and 0.22, respectively. Furthermore, Table 6 highlights notable differences in the positive predictive value (PPV) and sensitivity metrics, reinforcing the critical role of placing effective feature extraction, as facilitated by ResUDM, at the forefront of the model’s processing sequence to amplify the effectiveness of the subsequent stages handled by Mamba. The effectiveness of ResUDM heavily relies on the quality of feature representations generated by its Mamba component, which highlights the crucial impact of the sequential arrangement of these components on the model’s overall performance.

Integrating the convolutional block attention module (CBAM) into the ResUDM + Mamba and Mamba+ ResUDM architectures significantly enhances their segmentation performance, as evident through improvements to key metrics such as the dice coefficient, Hausdorff distance, positive predictive value (PPV), and sensitivity. This enhancement reflects the efficacy of CBAM in refining the segmentation accuracy and precision, which is critical for detailed tumor delineation. For the ResUDM + Mamba configuration, the addition of CBAM leads to an incremental improvement in the dice coefficients across whole tumor (WT), tumor core (TC), and enhancing tumor (ET) regions, as detailed in Table 7. This enhancement in segmentation accuracy is further corroborated by reductions in the Hausdorff distance for all tumor regions, indicating a closer alignment between the predicted and actual tumor boundaries, which is a testament to the precision CBAM offers.

The Mamba+ ResUDM architecture that includes CBAM shows noticeable improvements in dice scores for WT, TC, and ET segments, as highlighted in Table 8. The positive impact of CBAM extends to the Hausdorff distance measurements, where decreases across all tumor regions suggest more accurate boundary delineation.

The effectiveness of CBAM is not limited to accuracy and precision metrics alone. In the ResUDM + Mamba model, the integration of CBAM enhances both PPV and sensitivity across all tumor regions, indicating a refined precision–recall balance critical for effective segmentation. This improvement in diagnostic performance is evidenced in Table 9. The Mamba + ResUDM model exhibits similar enhancements, with increased PPV and sensitivity across tumor regions, demonstrating the module’s role in improving the model’s overall diagnostic capabilities, as shown in Table 10.

The consistent improvements across these diverse metrics underscore the pivotal role of CBAM for advancing the segmentation capabilities of brain tumor models. By meticulously analyzing the impact of CBAM, it is evident that the module boosts the models’ accuracy and precision and enhances their ability to accurately segment tumors, marking a significant advancement in medical imaging.

## 6. Conclusions

The MambaBTS model, as delineated in this study, amalgamates the robust framework of CNNs with the avant-garde Mamba structure, heralding a new era in the domain of brain image segmentation. Central to the ethos of MambaBTS is a dual-strategy design: the integration of cascade residual multi-scale convolutional kernels, the incorporation of the Mamba structure for advanced temporal feature handling, and the strategic implementation of a hybrid loss function that combines dice loss with cross-entropy. This fusion of methodologies refines the segmentation process and significantly reduces computational complexity while expanding the receptive field.

Employing cascade residual multi-scale convolutional kernels is instrumental for refining the segmentation process: it markedly reduces computational complexity while expanding the receptive field. This method does not merely maintain model efficiency and segmentation precision; it elevates them. By capturing features across diverse scales in cascade mode, MambaBTS achieves a nuanced comprehension of the input data, adeptly identifying intricate details alongside overarching patterns. Further, incorporating the Mamba structure within MambaBTS significantly augments the model’s proficiency at processing temporal features, which is a notable challenge for conventional CNN architectures and one that demands considerable computational resources in transformer-based models. This integration showcases the model’s enhanced capability to navigate the complexities inherent in temporal data analysis. Supplementary experiments provide a robust testament to the MambaBTS model’s superior performance. When benchmarked against established methodologies, MambaBTS demonstrates its efficacy across various evaluation metrics, affirming its status as a cutting-edge and efficacious approach to brain image segmentation.

Notably, the clinical implications of the MambaBTS model are profound. By significantly enhancing the speed and accuracy of brain image segmentation, this model facilitates earlier and more precise diagnoses, which are critical for managing neurological conditions. The ability to process temporal features with enhanced efficiency holds promise for monitoring disease progression and evaluating treatment efficacy in real time, offering a considerable advantage in personalized medicine. Future research endeavors will refine the model’s architecture to enhance its generalizability and practical utility. Additionally, assessing the model’s performance across a broader spectrum of medical imaging tasks constitutes a pivotal area of exploration. The overarching goal is to establish MambaBTS as a foundational tool for improving the precision and reliability of clinical diagnostics and treatments, ultimately contributing to tangible advancements in patient care.

## Figures and Tables

**Figure 1 entropy-26-00385-f001:**
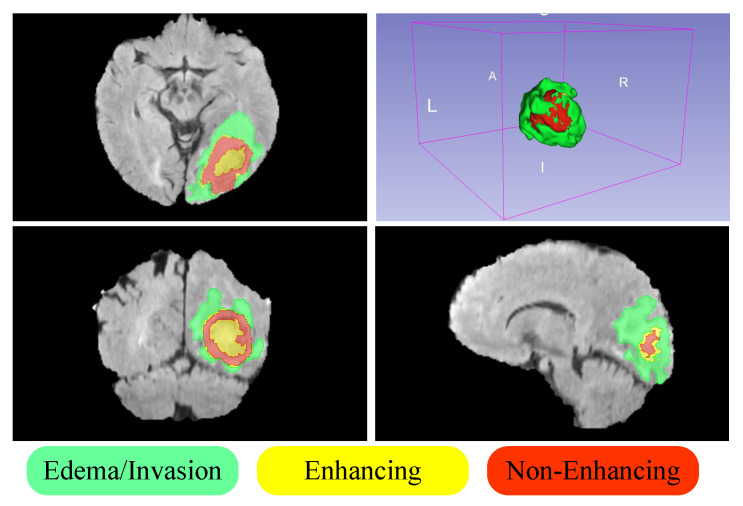
Green represents peritumoral edema (ED), yellow denotes enhancing tumor (ET), red signifies non-enhancing tumor (NET), and the background is depicted in black.

**Figure 2 entropy-26-00385-f002:**
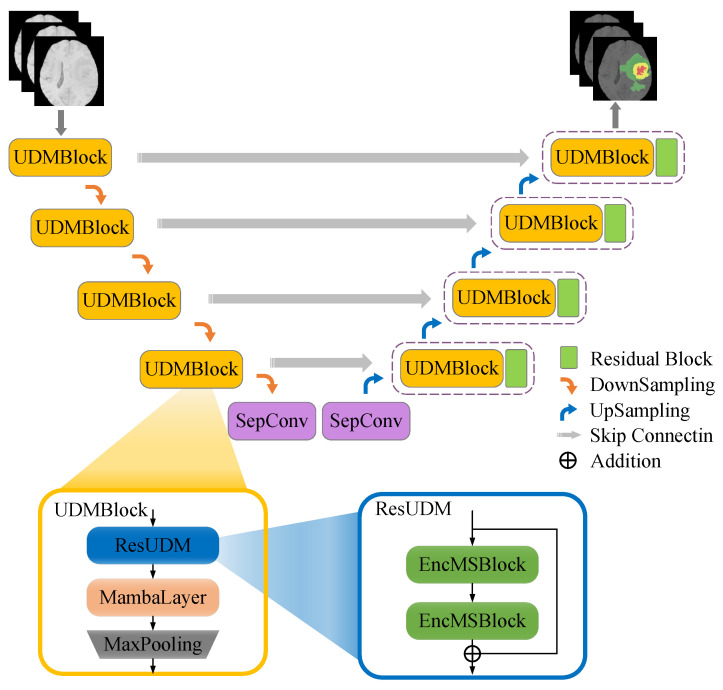
The overall architecture of the proposed MambaBTS.

**Figure 3 entropy-26-00385-f003:**
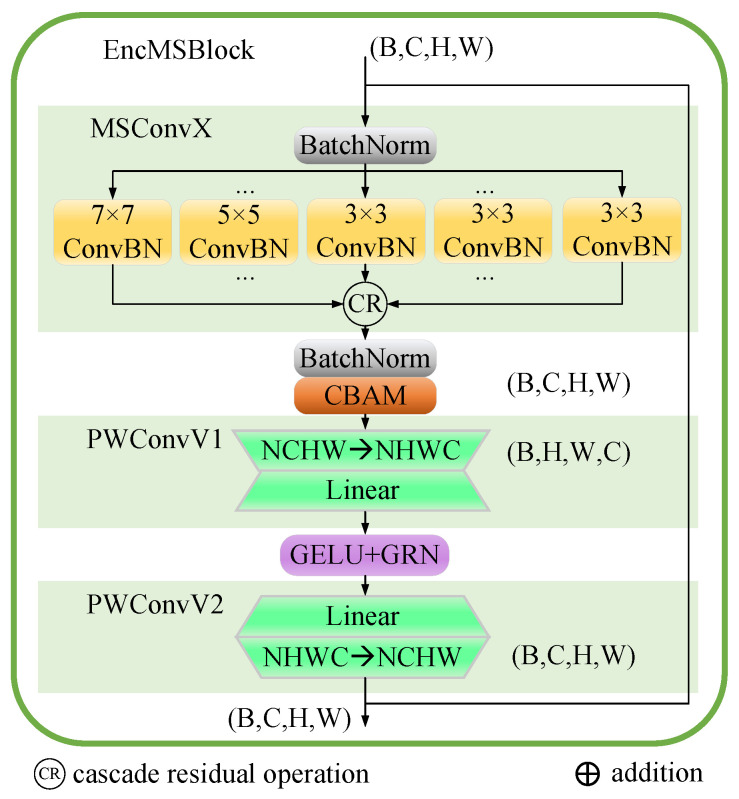
The architecture of EncMSBlock.

**Figure 4 entropy-26-00385-f004:**
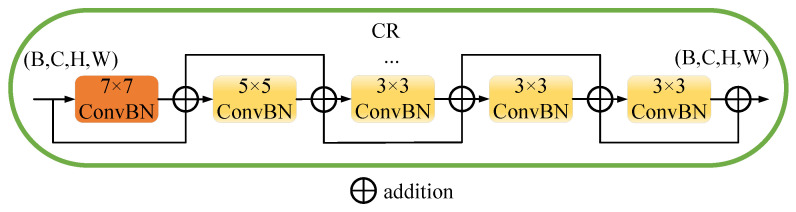
Sequential multi-scale ConvBN layers with residual connections in the CR, showcasing the flow from 7 × 7 to 5 × 5 to multiple 3 × 3 convolutions.

**Figure 5 entropy-26-00385-f005:**
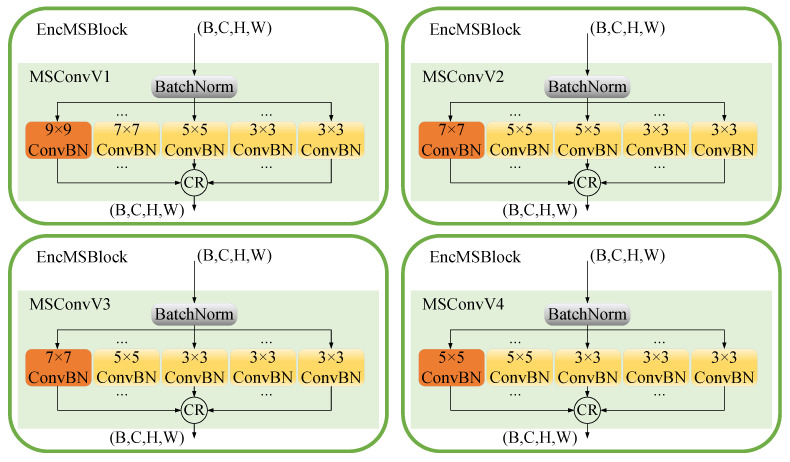
Four types of multi-scale convolutions.

**Figure 6 entropy-26-00385-f006:**
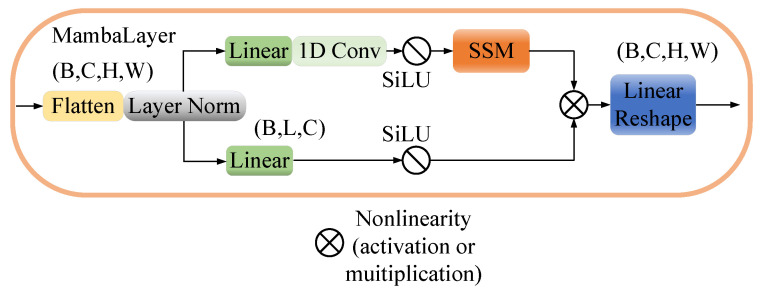
Schematic of MambaLayer with linear transformation.

**Figure 7 entropy-26-00385-f007:**
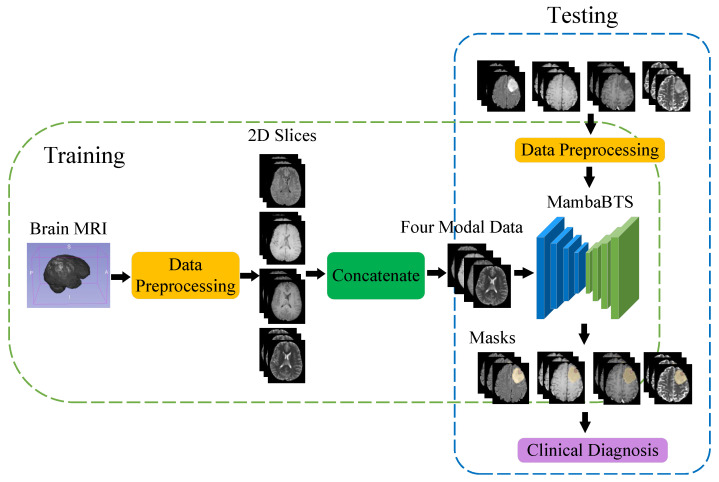
Comprehensive architecture flowchart for training and testing processes in brain tumor segmentation.

**Figure 8 entropy-26-00385-f008:**
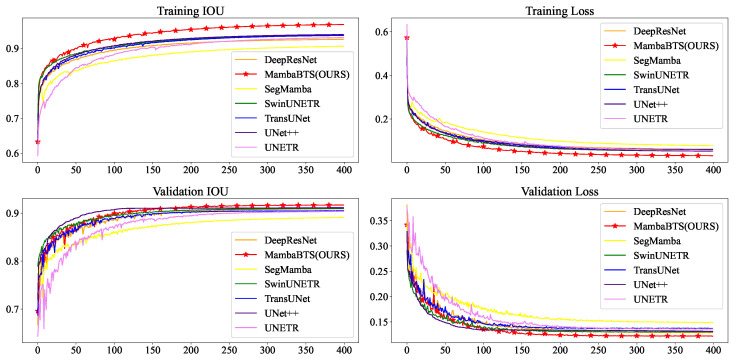
Comparative IoU and loss metrics across models on training and validation datasets.

**Figure 9 entropy-26-00385-f009:**
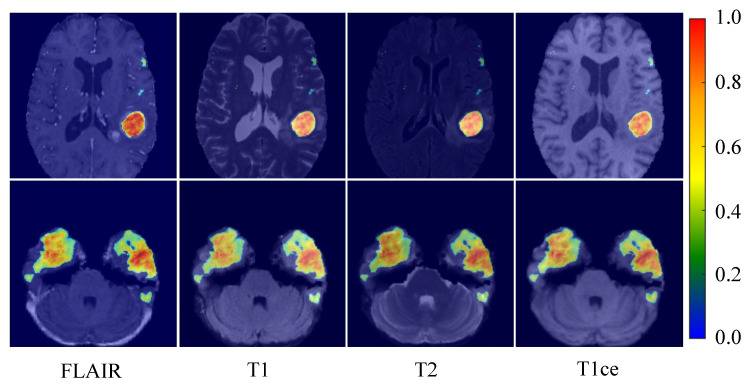
Heatmaps of the brain tumor segmentation model under different MRI modalities. In the heatmap, hot zones (from red to yellow) represent areas with a high probability of tumor presence per the model’s prediction. In contrast, cold zones (blue) indicate areas with a lower prediction probability.

**Figure 10 entropy-26-00385-f010:**
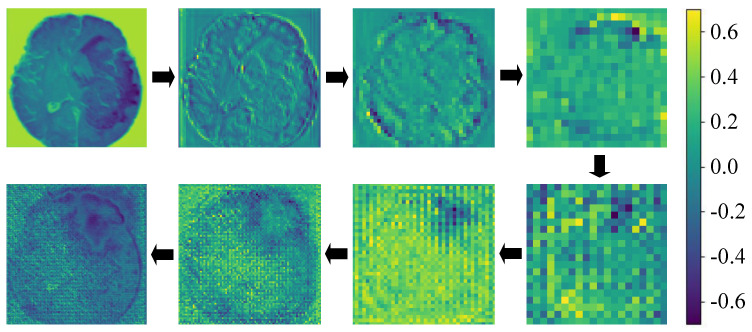
Visualization of feature maps in model’s feature extraction process. Different colors indicate the model’s focus areas, with deeper colors denoting higher attention weights.

**Figure 11 entropy-26-00385-f011:**
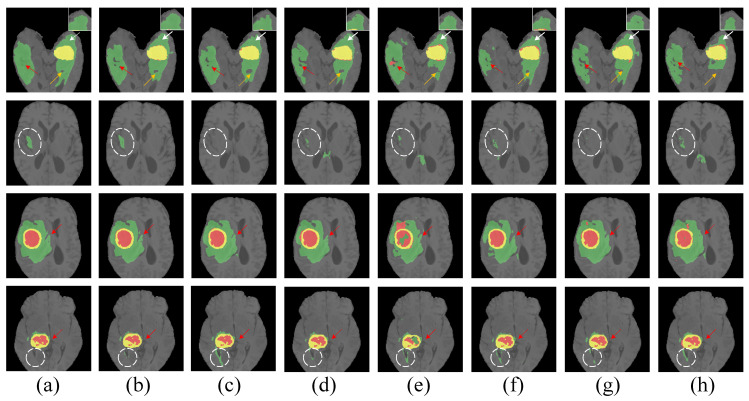
Visual comparison of brain tumor segmentation outcomes. (**a**) Ground truth, (**b**) MambaBTS (our model), (**c**) UNet++, (**d**) DeepResNet, (**e**) UNETR, (**f**) TransUNet, (**g**) Swin-UNETR, and (**h**) SegMamba mark the enhancing tumor (ET) in yellow, the tumor core (TC) in yellow and red, and the whole tumor (WT) in yellow, red, and green.

**Table 1 entropy-26-00385-t001:** Comparison of method parameters and FLOPs.

Method	Parameters	FLOPs
UNet++	36.63 M	54 G
DeepResNet	31.57 M	22 G
UNETR	95.39 M	27 G
TransUNet	105.21 M	14 G
Swin-UNETR	25.14 M	27 G
SegMamba	22.86 M	13 G
**Ours**	**18.09 M**	**8 G**

Note: FLOPs represent the number of floating-point operations. Parameters denote adjustable variables or weights within a model that were acquired during training to dictate its behavior and efficacy.

**Table 2 entropy-26-00385-t002:** Results of various algorithms on the BraTS 2019 validation set in terms of dice, PPV, and sensitivity metrics.

Method	Dice ↑			PPV ↑			Sensitivity ↑			Average ↑
	**WT**	**TC**	**ET**	**WT**	**TC**	**ET**	**WT**	**TC**	**ET**	
UNet++	0.8348	0.8308	0.7636	0.8498	0.8743	0.7717	0.8620	0.8928	0.8220	0.8335
DeepResNet	0.8372	0.8291	0.7662	**0.8653**	0.8536	0.7792	0.8539	0.9012	0.8023	0.8320
UNETR	0.7777	0.6988	0.6780	0.8296	0.7912	0.7040	0.7760	0.8057	0.7234	0.7538
TransUNet	0.8285	0.8475	0.7593	0.8612	0.8886	0.7873	0.8336	0.8928	0.7893	0.8320
SwinUNETR	0.8280	0.8332	0.7618	0.8456	0.8651	0.7764	0.8524	0.9030	0.8079	0.8304
SegMamba	0.8124	0.8093	0.7334	0.8269	0.8542	0.7390	0.8334	0.8804	0.7781	0.8075
**OURS**	**0.8450**	**0.8606**	**0.7796**	0.8597	**0.8920**	**0.7894**	**0.8716**	**0.9062**	**0.8305**	**0.8483**

**Table 3 entropy-26-00385-t003:** Results of various algorithms on the BraTS 2019 validation set in terms of BIoU.

Method	BIoU ↑			Average ↑
	**WT**	**TC**	**ET**	
UNet++	0.8551	0.7268	0.8053	0.7957
DeepResNet	0.8590	0.7238	0.8073	0.7967
UNETR	0.8026	0.6599	0.7364	0.7330
TransUNet	0.8537	0.7273	0.8053	0.7954
SwinUNETR	0.8520	0.7282	0.8050	0.7951
SegMamba	0.8332	0.7092	0.7783	0.7736
**OURS**	**0.8645**	**0.7350**	**0.8175**	**0.8057**

**Table 4 entropy-26-00385-t004:** Results of various algorithms on the BraTS 2019 validation set in terms of Hausdorff distance.

Method	Hausdorff ↓			Average ↓
	**WT**	**TC**	**ET**	
UNet++	2.6984	1.6660	2.8375	2.4006
DeepResNet	2.6641	1.7364	2.8118	2.4041
UNETR	2.9388	2.2502	3.2500	2.8130
TransUNet	2.6762	1.6302	2.8067	2.3710
SwinUNETR	2.6943	1.6855	2.8292	2.4030
SegMamba	2.7724	1.8312	2.9666	2.5234
**OURS**	**2.6511**	**1.6086**	**2.7813**	**2.3470**

**Table 5 entropy-26-00385-t005:** The impact of component sequence on dice score and Hausdorff distance.

Components			Dice ↑			Hausdorff ↓		
♣	♠	★	**WT**	**TC**	**ET**	**WT**	**TC**	**ET**
✓		✓	0.7915	0.8443	0.7145	2.9005	1.6245	3.0058
	✓	✓	**0.8450**	**0.8606**	**0.7796**	**2.6511**	**1.6086**	**2.7813**

♣: Mamba + ResUDM ♠: ResUDM + Mamba ★: CBAM.

**Table 6 entropy-26-00385-t006:** The impact of component sequence on PPV and sensitivity.

Components			PPV ↑			Sensitivity ↑		
♣	♠	★	**WT**	**TC**	**ET**	**WT**	**TC**	**ET**
✓		✓	0.8416	**0.9028**	0.7521	0.7993	0.8787	0.7494
	✓	✓	**0.8597**	0.8920	**0.7894**	**0.8716**	**0.9062**	**0.8305**

♣: Mamba + ResUDM ♠: ResUDM + Mamba ★: CBAM.

**Table 7 entropy-26-00385-t007:** Effect of CBAM on dice coefficient and Hausdorff distance in ResUDM+Mamba architecture.

Components		Dice ↑			Hausdorff ↓		
♠	★	**WT**	**TC**	**ET**	**WT**	**TC**	**ET**
✓	✓	**0.8450**	**0.8606**	**0.7796**	**2.6511**	**1.6086**	**2.7813**
✓		0.8448	0.8563	0.7766	2.6569	1.6105	2.8090

♠: ResUDM + Mamba ★: CBAM.

**Table 8 entropy-26-00385-t008:** Effect of CBAM on dice coefficient and Hausdorff distance in Mamba+ResUDM architecture.

Components		Dice ↑			Hausdorff ↓		
♣	★	**WT**	**TC**	**ET**	**WT**	**TC**	**ET**
✓	✓	**0.7915**	**0.8443**	**0.7145**	**2.9005**	**1.6245**	**3.0058**
✓		0.7823	0.8389	0.7039	2.9435	1.6672	3.0489

♣: Mamba + ResUDM ★: CBAM.

**Table 9 entropy-26-00385-t009:** Effect of CBAM on PPV coefficient and sensitivity in ResUDM+Mamba architecture.

Components		PPV ↑			Sensitivity ↑		
♠	★	**WT**	**TC**	**ET**	**WT**	**TC**	**ET**
✓	✓	**0.8597**	**0.8920**	**0.7894**	0.8716	0.9062	0.8305
✓		0.8535	0.8898	0.7765	**0.8768**	**0.9074**	**0.8393**

♠: ResUDM + Mamba ★: CBAM.

**Table 10 entropy-26-00385-t010:** Effect of CBAM on PPV coefficient and sensitivity in Mamba+ResUDM architecture.

Components		PPV ↑			Sensitivity ↑		
♣	★	**WT**	**TC**	**ET**	**WT**	**TC**	**ET**
✓	✓	**0.8416**	**0.9028**	**0.7521**	**0.7993**	0.8787	**0.7494**
✓		0.8326	0.8828	0.7470	0.7908	**0.8901**	0.7323

♣: Mamba + ResUDM ★: CBAM.

## Data Availability

The datasets analyzed for this study are derived from the Multi-modal Brain Tumor Segmentation Challenge 2019, as described in the following reference: S. Bakas et al., “Multi-modal Brain Tumor Segmentation Challenge 2019”, Center for Biomedical Image Computing and Analytics, Perelman School of Medicine at the University of Pennsylvania, 2019. The Center for Biomedical Image Computing and Analytics, Perelman School of Medicine at the University of Pennsylvania, provides public access to these datasets through their website at https://www.med.upenn.edu/cbica/brats2019.html (accessed on 26 March 2024).

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
