# Peer review of "Cascade Residual Multiscale Convolution and Mamba-Structured UNet for Advanced Brain Tumor Image Segmentation"

_entropy, 2024, doi:10.3390/e26050385_

Round 1
Reviewer 1 Report
Comments and Suggestions for Authors
This manuscript presents an innovative approach to brain tumor segmentation, combining the strengths of Convolutional Neural Networks and Transformers through the introduction of a MambaBTS model. The attempt to address the computational inefficiencies and segmentation accuracy challenges in current models is commendable. However, there are several areas where the paper could be improved to enhance clarity, contribution significance, and experimental validation:
Q1. The manuscript would benefit significantly from including figures that clearly delineate the classes involved in the study, specifically the definitions and distinctions among the whole tumor, tumor core, and enhancing tumor. This clarification will aid readers in better understanding the segmentation challenges and the dataset used.
Q2. Additionally, the schematic diagrams and figures within the paper contain text that is too small to read comfortably, reducing their usefulness. Enhancing the legibility of these figures by increasing text size and contrast could significantly improve the paper's clarity and effectiveness as a communication tool.
Q3. While the introduction of the MambaLayer and the use of a combined BCE and Dice loss function are noted, the paper lacks a clear articulation of its unique contributions to the field. A more detailed discussion comparing the proposed model to existing methodologies, highlighting the specific advancements and innovations brought by this study, would greatly reinforce the paper's significance.
Q4. The manuscript currently lacks a comprehensive description or illustration of the proposed algorithm's main framework. Including a step-by-step breakdown or a flowchart of the MambaBTS model, from input to final segmentation, would offer readers a clearer understanding of the methodology and enhance reproducibility.
Q5. The application of Intersection over Union (IoU) as an evaluation metric for segmentation problems is standard. However, for a nuanced task such as instance segmentation of brain tumors, it may be more appropriate to consider metrics like Boundary IoU (BIoU) and mean Average Precision (mAP). These metrics could offer a more detailed assessment of the model's performance in delineating tumor boundaries and distinguishing between tumor instances. A discussion regarding the choice of evaluation metrics, including their advantages and potential limitations for this specific application, would enrich the paper.
Reviewer 2 Report
Comments and Suggestions for Authors
In this paper the authors aim to introduces MambaBTS, a model, derivated from Convolutional Neural Networks and Transformers inspired by the Mamba architecture, for brain tumor imaging segmentation. Their results underscore the model’s potential to offer a balanced, efficient, and effective segmentation method.
The study is potentially interesting, the methods are sounding to me, the work seems well performed and sufficiently detailed, the figures and tables are clear.
Some points remain to be clarified:
In Introduction (line 18-21): I don't understand why the authors introduce brain cancer by talking about the psychological aspects and well-being. I believe that this introduction needs to be radically changed because it does not capture the salient aspects of the disease. Also also in relation to a greater importance of tumor segmentation.
Line 359: For a better functioning of the segmentation, is it preferable to choose one sequence over another? Does the same sequence with different parameters have different segmentation results?
Line 477: In my opinion, in the conclusions, the advantages of this method from a clinical point of view should be underlined to give greater value to the manuscript.
Comments on the Quality of English LanguageCan be improved.
